# Encoding off-shell effects in top pair production in Direct Diffusion networks

**Mathias Kuschick**[1†]

**1** University of Münster

† mathias.kuschick@uni-muenster.de

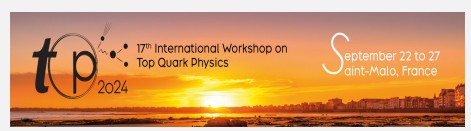

## Abstract

**To meet the precision targets of upcoming LHC runs in the simulation of top pair production events it is essential to also consider off-shell effects. Due to their great computational cost I propose to encode them in neural networks. For that I use a combination of neural networks that take events with approximate off-shell effects and transform them into events that match those obtained with full off-shell calculations. This was shown to work reliably and efficiently at leading order. Here I discuss first steps extending this method to include higher order effects.**

## 1 Introduction

To contrast experimental measurements at the Large Hadron Collider (LHC) with theoretical predictions precise theoretical simulations are essential. Simulating the production and the decay of heavy particles and restricting their kinematics to mass shell, even approximately, can however introduce bias. As the simulation of processes with full off-shell kinematics is computationally very costly I propose the use of neural network surrogates to encode the off-shell effects. This approach, applied to top pair production with dileptonic decay at leading order (LO) in QCD in Ref. [1], uses a Bayesian Direct Diffusion network to map the events simulated with approximate off-shell kinematics to those obtained with full off-shell calculations. This method, that transforms the event kinematics, is chosen over the alternative in which events are generated to ensure that the neural network has to learn the least adjustment possible. Additionally, a classifier neural network is used to reweight the transformed events to further align them with the target sample. In this follow-up work I demonstrate the viability of this approach at next-to-leading order (NLO) in QCD. For more details I refer to the original study.

It must be mentioned at this point that our consideration of the NLO in the proceedings is only the first step. In particular I only concern myself with real radiation in the production process. This guarantees that the samples I transform into each other have the same number of particles in the final state, which is crucial for our neural network setup.

## 2 Data

In the original study the reference process is the top pair production with leptonic decays at LO. Here I chose the same process as reference but this time at NLO. The event generators HVQ [2] and ST-WTCH-DR [3] were used to generate doubly- and singly-resonant events with approximate off-shell effects, respectively. For the sake of notational simplicity I refer to these events as on-shell events. The generator BB4L [4, 5] was used to generate events in a fully off-shell calculation. In the following I call these events off-shell events. All the events were generated with the same physical parameters as shown in our original study.

The ST-WTCH-DR generator implements the $tW$ associated production in 5 flavour-number-scheme and is thus missing one particle as compared to the HVQ generator. To rectify this I use PYTHIA [6] to attach just one gluon or light quark in the production process (initial- or final-state) in shower approximation. Thus the on-shell events all have the same number of final state particles, or dimensionality.

At NLO both input (on-shell) and target (off-shell) events now contain additional particles as compared to LO. BB4L events contain up to two additional particles

$$
\left.
\begin{aligned}
pp &\to b e^+ \nu_e \, \bar{b} \mu^- \nu_\mu \\
pp &\to b e^+ \nu_e \, \bar{b} \mu^- \nu_\mu \, j
\end{aligned}
\right\} \qquad \text{``on-shell''} \tag{1}
$$

$$
\left.
\begin{aligned}
pp &\to b e^+ \nu_e \, \bar{b} \mu^- \nu_\mu \\
pp &\to b e^+ \nu_e \, \bar{b} \mu^- \nu_\mu \, j \\
pp &\to b e^+ \nu_e \, \bar{b} \mu^- \nu_\mu \, j j' \\
pp &\to b e^+ \nu_e \, \bar{b} \mu^- \nu_\mu \, j j' j'
\end{aligned}
\right\} \qquad \text{``off-shell''} \tag{2}
$$

Here, the massless quarks or gluons $j$ are attached in the production process and the gluons $j'$ are attached in the decay process. A comparison between the two event distributions on the example of the reconstructed top mass and the mass of lepton–$b$-jet system can be seen in Fig. 1.

Since I concern myself here only with real radiation in the production process, the gluons $j'$ need to be removed from the off-shell events to match their dimensionality with the on-shell events. To preserve the correct top quark kinematics I use the fact that BB4L optionally provides the events' Born kinematics. Lorentz-boosting the final state particles underlying Born momenta $p_{i,\text{born}}$ into the inertial frame of the respective top or antitop quark according to Eq. 3 transforms the events to a state where the gluons are not yet emitted but their summed momenta already describe the correct top or antitop quark kinematics after a gluons emission

$$
\begin{aligned}
p_i' &= \Lambda(-v_t)\Lambda(v_{t,\text{born}})p_{i,\text{born}} && \text{for } i \in \big(b, e^+, \nu_e\big) \\
p_i' &= \Lambda(-v_{\bar{t}})\Lambda(v_{\bar{t},\text{born}})p_{i,\text{born}} && \text{for } i \in \big(\bar{b}, \mu^-, \bar{\nu}_\mu\big).
\end{aligned} \tag{3}
$$

The redistribution of the final state particle momenta after reattaching the gluons can be done later without the need of machine learning techniques.

In Eq. 3 $\Lambda$ refers to the Lorentz transformation. The velocities $v$ are either top or antitop quark velocities as described in the subscript, where velocities in the Born approximation are marked as such. Further, before treating the data with neural networks, the dimensions of the phase space are reduced by transforming the momentum components $(E, p_x, p_y, p_z)$ of each final state particle to $(p_T, \phi, \eta, m)$ where the mass $m$ is constant due to the particle being on the mass shell. It can therefore be omitted. I rotate all azimuthal angles $\phi$ so that $\phi_{\bar{\nu}} = 0$ for all events. As two dimensions can be omitted to satisfy

$$
\sum_i p_{T,i} = 0 \qquad \text{for } i \in \big(b, e^+, \nu_e, \bar{b}, \mu^-, \bar{\nu}_\mu, j\big), \tag{4}
$$

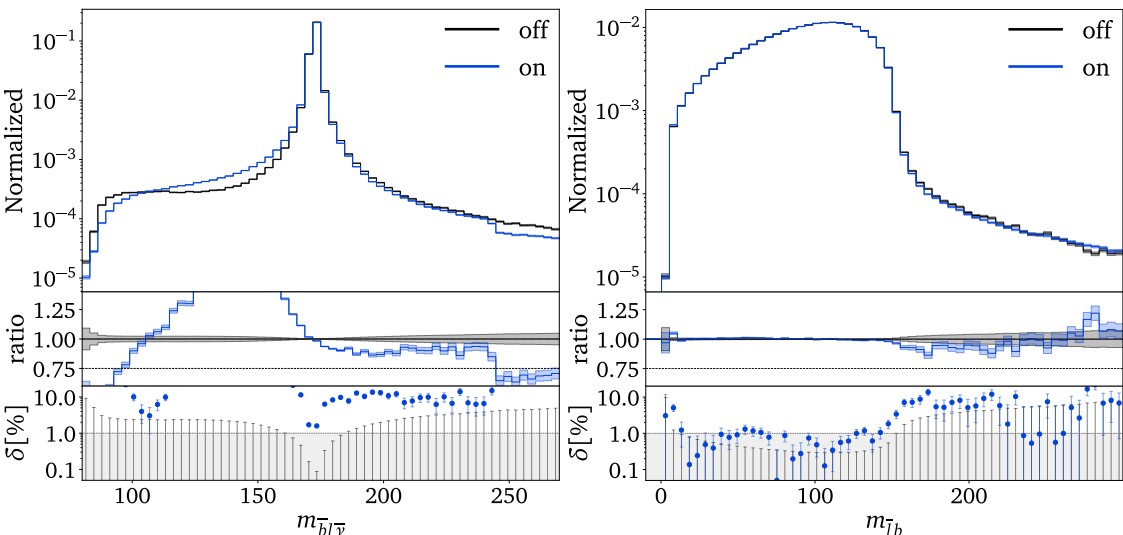

Figure 1: Exemplary observable distributions of the reconstructed antitop quark mass $m_{\bar{b}l\bar{\nu}}$ as well as $m_{lb}$. The on-shell distribution consists of the summed up distributions of on-shell doubly-resonant events as well as singly-resonant events. The discontinuity in the $m_{\bar{b}l\bar{\nu}}$ distribution seen on the left side is due to approximating the off-shell calculations for doubly-resonant events using finite top width.

I reduce the initial 28 dimensions to 18 dimensions in total.

## 3 The Bayesian Direct Diffusion network

A Bayesian Direct Diffusion network is used to map events in the on-shell phase space to the off-shell phase space. It iteratively learns how to predict a velocity field between the on-shell and the off-shell distributions spanning along an additional unphysical dimension. Single events can be transported from the on-shell distribution to their target places in the off-shell distribution by solving the differential equation given by that velocity field. What happens each iteration of the training process is shown in the diagram in Fig. 2 below.

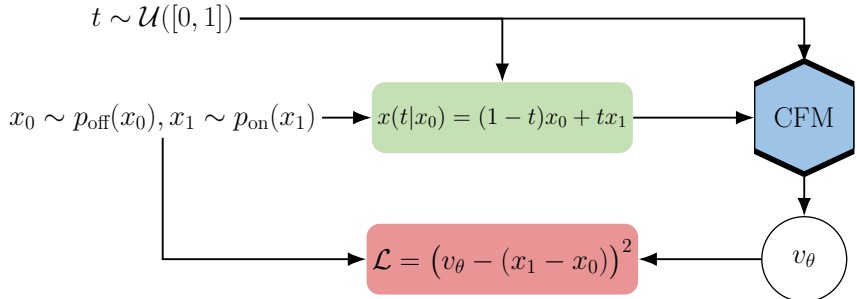

Figure 2: Schematic showing what happens each iteration of training the CFM network. Diagram adapted from Ref [7].

This training procedure is also called conditional flow matching (CFM) [7,8]. The network predicts a velocity $v_\theta$ for a random point on the linear trajectory that connects two randomly sampled events of each distribution. Then it tries to minimize the mean squared error between prediction and correct $v(t|x_0)$. It can be shown that this seemingly random learning of linear

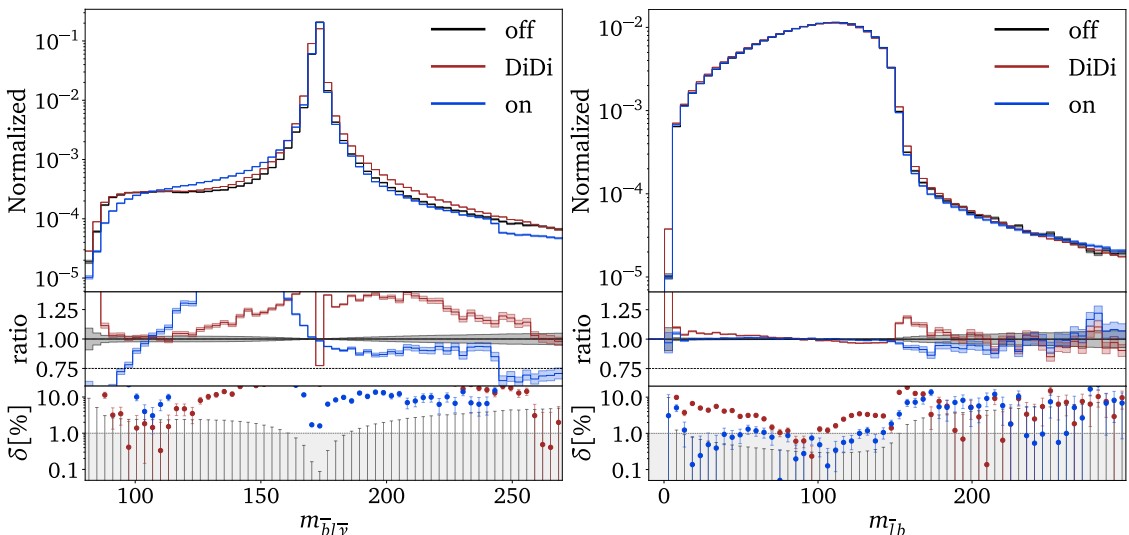

Figure 3: Exemplary results of the Direct Diffusion (DiDi) network.

trajectories results in an average velocity field that correctly maps between both phase spaces. The hyperparameters used in the training can be seen in Tab. 1.

The results of this transformation of events can be seen in the plots shown in Fig. 3. One might think that it is advantageous to deal with singly-resonant and doubly-resonant events separately than to let one network learn the transformation of both types of events jointly. However, any advantage due to this separation has shown to not matter much at the Les Houches Event (LHE) stage. Nevertheless, it could be shown that the use of singly and doubly-resonant events as input for the network provides better results than the use of only doubly-resonant on-shell events, which shows the importance of the quality of the input event distribution. For the results shown here I decided to train a single network. Each training batch of top pair events is mixed with a certain percentage of randomly sampled single top events equal to their proportion in the events generated with BB4L. The percentages are 2.7506% for antitops and 2.7477% for tops.

The Direct Diffusion network described here can be realized as a normal feedforward neural network. That means the learnable variables of the network are weights and biases represented by numbers. I decided to implement the network as a Bayesian neural network [9, 10] where biases and weights are not represented by numbers but by normal distributions with a learnable mean and standard deviation. This enables us to sample from a wide range of possible trained networks. Using a set of sampled networks I transform each input event into a set of generated events. The variance in the individual variables of the resulting event kinematics provides information about the uncertainty of the network's prediction, which can be approximated as a normal distribution fit to the generated kinematics in each dimension.

| Hyperparameter | DiDi | classifier |
|---|---|---|
| Embedding dimension | 64 | |
| Layers | 8 | 5 |
| Intermediate dimensions | 1024 | 512 |
| Dropout | | 0.1 |
| Normalization | | BatchNorm1d |
| LR scheduling | OneCycle | ReduceOnPlateau |
| Starter LR | $10^{-4}$ | $1^{-3}$ |
| Max LR | $10^{-3}$ | |
| Patience | | 10 |
| Epochs | 1000 | 100 |
| Batch size | 65536 | 16384 |
| $c$ | $10^{-4}$ | |
| # Training events $(t\bar{t}, t, \bar{t})$ | 2.4 M, 731 K, 732 K | 2 M, 610 K, 610 K |

Table 1: Hyperparameters used in the Direct Diffusion and classifier network trainings.

## 4    Classifier based reweighting

A classifier network is used to further improve the distribution of the generated event kinematics. The classifier network is trained to differentiate between both types of events, generated events and true off-shell events, which are labeled as such. The network should output a 1 for off-shell events and a 0 for generated events. Since the network is trained to minimize the mean squared error between prediction an correct event label, this leads to the implicit estimation of densities of events in the phase space.

$$C(x) = \frac{p_{\text{off,data}}(x)}{p_{\text{off,data}}(x) + p_{\text{off,model}}(x)} \tag{5}$$

$$w(x) = \frac{p_{\text{off,data}}(x)}{p_{\text{off,model}}(x)} = \frac{C(x)}{1 - C(x)} \tag{6}$$

This can be seen in Eq. 5, where $C(x)$ is the output of the classifier between 0 and 1 and $x$ is a arbitrary point in the phase space. It can be used to describe the weight $w(x)$ as shown in Eq. 6. Given events represented by points in phase space $x$ and generated by the Direct Diffusion network the classifier is then used to reweight every event which results in a generated event distribution that resembles the target off-shell distribution more closely [10]. The hyperparameters used for the training of the classifier can be seen in Tab. 1. Its results can be seen Fig. 4. One can see that the reweighting has a appreciable effect on the generated event distribution and brings it closer to the desired off-shell distribution. Here, it should be mentioned that the reweighting step does not work properly without the first step of transforming the data using the Direct Diffusion network. That is because huge parts of the phase space are not populated in the on-shell data set and therefore cannot be reweighted.

## 5    Conclusion

It was succesfully demonstrated that the transformation from on-shell to off-shell event distributions can also be realised in the higher-dimensional NLO phase space compared to the LO

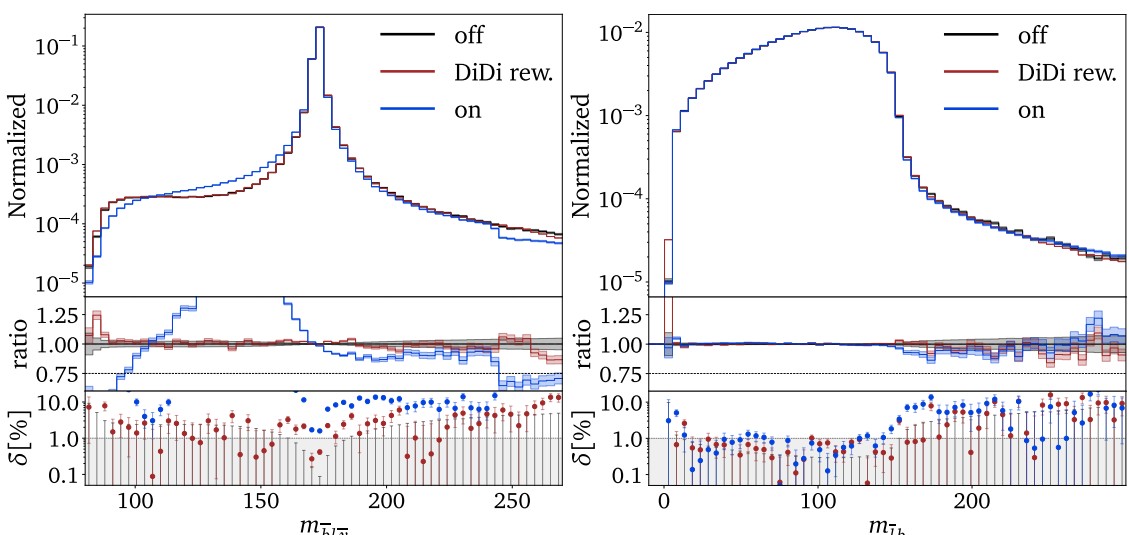

Figure 4: Results of the Direct Diffusion network after additional reweighting by the classfier network.

phase space. The combination of Direct Diffusion and classifier network makes it possible to generate even complicated off-shell event distributions that match the actual off-shell distribution with a deviation of just a few percent. Over large regions of the phase space, the deviation even falls below the one percent mark. For the complete generation of off-shell events at NLO, it is necessary to reattach the previously removed radiation in decay mentioned in section 2. I will deal with this topic in a future paper. In this paper I will also subject the results to more detailed tests, e.g. comparing the showered events. I will also discuss the ways in which the networks can be implemented.

## Acknowledgements

The work of MK is supported by the BMBF through the project InterKIWWU. I thank Anja Butter, Tomáš Ježo, Michael Klasen, Sofia Palacios Schweitzer and Tilman Plehn for the collaboration on the original study and fruitful discussion in this follow-up work.

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
