# Peer review of "Encoding off-shell effects in top pair production in Direct Diffusion networks"

_SciPost Physics Proceedings_

## Round 2 · Referee Report · Anonymous (Referee 1) · 2025-2-13

Report

The writeup is a follow-up of a published paper, but it includes new results. I have only a few questions/requests: - the list of references is a little short and a little centered around the author group of the original paper. Please expand, including a comment on Schrodinger bridges, which are similar to direct diffusion. - reading the proceedings, I see that it goes beyond the original paper, but could the authors be more specific what is new and what changes under physics and technical aspects? - is the classifier also Bayesian? Would it make sense to train it that way or is the error from it sub-leading for instance in Fig.4.

Recommendation

Ask for minor revision

---

## Round 3 · Referee Report · Tilman Plehn (Referee 1) · 2025-3-18

Report

Thank you for considering my comments, I am happy.

Recommendation

Publish (surpasses expectations and criteria for this Journal; among top 10%)

---

## Round 3 · List of Changes

• a paragraph on Schrödinger bridges was added
  • a paragraph further describing the changes in the sample distributions in comparison to the original study was added
  • I added the information that the classifier is not yet bayesianized

---

## Editorial Decision

accepted_in_target_journal